# The Interplay among Glucocorticoid Therapy, Platelet-Activating Factor and Endocannabinoid Release Influences the Inflammatory Response to COVID-19

**DOI:** 10.3390/v15020573

**Published:** 2023-02-19

**Authors:** Jonatan C. S. de Carvalho, Pedro V. da Silva-Neto, Diana M. Toro, Carlos A. Fuzo, Viviani Nardini, Vinícius E. Pimentel, Malena M. Pérez, Thais F. C. Fraga-Silva, Camilla N. S. Oliveira, Augusto M. Degiovani, Fátima M. Ostini, Marley R. Feitosa, Rogerio S. Parra, José J. R. da Rocha, Omar Feres, Fernando C. Vilar, Gilberto G. Gaspar, Isabel K. F. M. Santos, Ana P. M. Fernandes, Sandra R. Maruyama, Elisa M. S. Russo, Vânia L. D. Bonato, Cristina R. B. Cardoso, Marcelo Dias-Baruffi, Lúcia H. Faccioli, Carlos A. Sorgi

**Affiliations:** 1Departamento de Química, Faculdade de Filosofia, Ciências e Letras de Ribeirão Preto-FFCLRP, Universidade de São Paulo-USP, Ribeirao Preto 14040-901, SP, Brazil; 2Departamento de Análises Clínicas, Toxicológicas e Bromatológicas, Faculdade de Ciências Farmacêuticas de Ribeirão Preto-FCFRP, Universidade de São Paulo-USP, Ribeirao Preto 14040-903, SP, Brazil; 3Programa de Pós-Graduação em Imunologia Básica e Aplicada-PPGIBA, Instituto de Ciências Biológicas, Universidade Federal do Amazonas-UFAM, Manaus 69080-900, AM, Brazil; 4Departamento de Bioquímica e Imunologia, Faculdade de Medicina de Ribeirão Preto-FMRP, Universidade de São Paulo-USP, Ribeirao Preto 14040-900, SP, Brazil; 5Hospital Santa Casa de Misericórdia de Ribeirão Preto, Ribeirao Preto 14085-000, SP, Brazil; 6Departamento de Cirurgia e Anatomia, Faculdade de Medicina de Ribeirão Preto-FMRP, Universidade de São Paulo-USP, Ribeirao Preto 14048-900, SP, Brazil; 7Departamento de Clínica Médica, Faculdade de Medicina de Ribeirão Preto-FMRP, Universidade de São Paulo-USP, Ribeirao Preto 14049-900, SP, Brazil; 8Departamento de Enfermagem Geral e Especializada, Escola de Enfermagem de Ribeirão Preto-EERP, Universidade de São Paulo-USP, Ribeirao Preto 14040-902, SP, Brazil; 9Departamento de Genética e Evolução, Centro de Ciências Biológicas e da Saúde, Universidade Federal de São Carlos-UFSCar, Sao Carlos 13565-905, SP, Brazil

**Keywords:** COVID-19, glucocorticoid, endocannabinoid, platelet-activating factor, inflammation

## Abstract

COVID-19 is associated with a dysregulated immune response. Currently, several medicines are licensed for the treatment of this disease. Due to their significant role in inhibiting pro-inflammatory cytokines and lipid mediators, glucocorticoids (GCs) have attracted a great deal of attention. Similarly, the endocannabinoid (eCB) system regulates various physiological processes including the immunological response. Additionally, during inflammatory and thrombotic processes, phospholipids from cell membranes are cleaved to produce platelet-activating factor (PAF), another lipid mediator. Nonetheless, the effect of GCs on this lipid pathway during COVID-19 therapy is still unknown. This is a cross-sectional study involving COVID-19 patients (*n* = 200) and healthy controls (*n* = 35). Target tandem mass spectrometry of plasma lipid mediators demonstrated that COVID-19 severity affected eCBs and PAF synthesis. This increased synthesis of eCB was adversely linked with systemic inflammatory markers IL-6 and sTREM-1 levels and neutrophil counts. The use of GCs altered these lipid pathways by reducing PAF and increasing 2-AG production. Corroborating this, transcriptome analysis of GC-treated patients blood leukocytes showed differential modulation of monoacylglycerol lipase and phospholipase A2 gene expression. Altogether, these findings offer a breakthrough in our understanding of COVID-19 pathophysiology, indicating that GCs may promote additional protective pharmacological effects by influencing the eCB and PAF pathways involved in the disease course.

## 1. Introduction

Severe acute respiratory syndrome coronavirus 2 (SARS-CoV-2) and its variants are the cause of coronavirus disease 2019 (COVID-19). This infectious disease is responsible for millions of fatalities and has prompted significant efforts all around the world to develop potential therapy options. The results of the disease range from asymptomatic infection to severe symptoms, acute respiratory distress syndrome (ARDS) and fatal multi-organ failure [1,2]. The development of systemic inflammation is caused by the activation of a large number of components, including cells and soluble substances, of the innate and adaptive responses. Depending on the produced amount, the cytokines and chemokines may induce a dysregulated inflammatory response, leading to multiple organ failure and a worse prognosis [3,4].

Despite their importance as an energy source and a component of cell membranes, lipids are involved in a wide variety of pathophysiological processes [5]. During infections, innate immune cells, such as granulocytes and monocytes/macrophages, are driven to sites of injury and produce bioactive lipids [6]. In the context of microbial or viral infection, polyunsaturated fatty acids (PUFAs) and their metabolites play a pivotal role. A key lipid mediator implicated in COVID-19 pathology is platelet-activating factor (PAF) [7]. More specifically, PAF is a glyceryl-ether phospholipid (1-O-alkyl-2-acetyl-sn-glycero-3-phosphocholine), which is an effective mediator of inflammation and thrombosis [8]. PAF-induced platelet aggregation and the activity of its metabolic enzymes are linked to a number of clinical conditions, such as asthma, atherosclerosis, heart failure, cancer and viral diseases [9,10]. Platelets, endothelial cells, macrophages, monocytes, neutrophils and other cells produce it constantly or in response to inflammatory stimuli [8]. Further, an increase in the activity of lipid metabolizing enzymes could lead to enhance the formation of lipid mediators belonging to the family of endocannabinoids (eCB) [11]. These lipid mediators have a role in the establishment of an efficient innate immune response during inflammation, such as immunosuppression during sepsis [11]. In several sepsis models, it has been observed that specific CB2R agonists, such as eCB, reduce leukocyte recruitment, oxidative burst, systemic inflammatory mediator production, bacteremia and lung tissue damage, while enhancing survival [12]. SARS-CoV-2 infections are linked to inflammation; however, activation of the eCB system may help minimize that [13,14,15].

Drugs that have been first used in the management of COVID-19 include antibiotics, heparin to manage thromboembolic events, antiviral agents, non-steroidal anti-inflammatory drugs (NSAIDs), antimalarials, interferon (IFN) type I and II, inhibitors of cytokines or of their receptors and glucocorticoids (GCs) [16,17,18,19,20,21]. For instance, GCs reduce cytokine production and inhibit the phospholipase A2 enzyme by regulating gene transcription and blocking the negative regulation of genes involved in inducible nitric oxide synthase [22]. The majority of their therapeutic actions result from their interaction with glucocorticoid receptors, which alters complex molecular systems including gene expression and intracellular signaling modulation. This results in the control of immune cell activation and function as well as the suppression of the production of several pro-inflammatory cytokines [23]. Moreover, this series of alterations leads to a reduction in the synthesis of lipid mediators, such as prostaglandins and leukotrienes, which are essential substances in the inflammatory process, thereby revealing a potential alternative pathway for the regulation of the expression of several immune components and resulting in anti-inflammatory effects [22,23].

There is currently a dearth of experimental evidence demonstrating the direct effect of COVID-19 on modifications to the PAF and eCB systems, as well as the possible repercussions of these adjustments. These lipid mediators regulate viral replication and influence the innate and adaptive immune responses of the host [24]. Importantly, the use of GCs in severe COVID-19 reduces mortality, but their risk/benefit balance remains inconclusive; the period and duration of treatment, illness phase, bacteria co-infections and co-morbidities may influence the efficacy or adverse effects of GCs [25]. In this work, biochemical and molecular investigation of plasma and blood cells from COVID-19 patients indicated a GC-induced increase in the levels of the eCB, 2-arachidonoylglycerol (2-AG), while also revealing a decrease in PAF generation by modification of lipid metabolism-enzyme gene expression. Regardless of the COVID-19 severity symptoms, this occurrence was found to be connected with the anti-inflammatory pathway that was caused by the therapy with GCs.

## 2. Materials and Methods

### 2.1. Study Design and Participants

This is an observational, descriptive, cross-sectional study, conducted in Brazil from June 2020 to January 2021, prior to the introduction of COVID-19 vaccination. Included in this study were 200 patients with positive RT-PCR nasopharyngeal swab results (Biomol OneStep Kit/COVID-19, Instituto de Molecular Biology of Paraná, IBMP Curitiba/PR, Brazil) or serological assays to detect anti–SARS-CoV-2 IgM/IgG/IgA (SARS-CoV-2 antibody test^®^, Guangzhou Wondfo Biotech Co., Ltd., Guangzhou, China) and 35 healthy control subjects who tested negative for SARS-CoV-2. This study was carried out in collaboration with Hospital Santa Casa de Misericórdia de Ribeirão Preto and Hospital São Paulo de Ribeirão Preto, Brazil, using the exclusion criteria: minors under 18, women who are pregnant or breastfeeding and individuals using immunomodulators. At the time of sample collection, patients were classified as mild/moderate (*n* = 55) or severe/critical (*n* = 145) symptoms based on a modified version of the Novel Coronavirus Pneumonia Diagnosis and Treatment Guideline, 7th Edition statement [26,27]. In addition to the conventional pharmaceutical standard of care, hospitalized patients received oral or intravenous dexamethasone, at a dose of 6 mg once daily, for up to 10 days or until hospital release, if sooner [28]. At the Supera Park-Scientific and Technological Development Center in Ribeirao Preto, Sao Paulo, Brazil, blood samples from healthy controls were taken. All participants provided written consent for the collection of samples and subsequent analysis.

### 2.2. Plasma Isolation

Peripheral blood samples were collected via venipuncture in tubes with a vacuum collection system within the first 24 h of patient enrollment to the study. Two tubes (5 mL capacity) were collected from each patient: one tube containing EDTA anticoagulant (BD Vacutainer^®^ EDTA K2, New Jersey, NJ, USA) to perform hematological tests and one tube containing heparin anticoagulant (BD SST^®^ Gel Advance^®^, New Jersey, NJ, USA). Plasma was separated by centrifugation at 400 g for 10 min at 4 °C and used to quantify levels of circulating mediators by ELISA assay, aliquots were stored with an equal volume of methanol (1:1 *v*/*v*) at −80 °C for further analysis of lipid mediators.

### 2.3. Sample Preparation and Extraction of eCBs and PAF

We used plasma samples (200 μL) from COVID-19 patients and controls. The approach for extracting samples was based on the Bligh and Dyer method [29]. In brief, methanol (MeOH; 500 μL) was added to the plasma sample and spiked with 10 μL of internal standard solution (IS; PAF C16-d4, PAF C18-d4, Lyso-PAF C16:d4, Lyso-PE 17:1, 2AG-d5 and AEA-d4 (1 μg/mL)—Cayman Chemicals, Ann Arbor, MI, USA; and Avanti Polar Lipids, Alabaster, AL, USA). Samples were then homogenized for 3 min. at 30 Hz (Bead Ruptor 96, Omni Inc., Kennesaw, GA, USA), chloroform (CHCl_3_; 250 μL) was added to the solution and it was homogenized for another 3 min. at 30 Hz. The entire amount was transferred to the glass tubes containing 250 μL of CHCl_3_ and 500 μL of milliQ-H_2_O. Subsequently, samples were vortexed for 10 min. and centrifuged at 1500× *g* for 10 min. at 4 °C and the entire volume of the lower phase (organic phase) was recovered. The upper phase was extracted with 250 μL of CHCl_3_ once more and the entire lower phase was recollected. The lower stages were mixed and dried for 1 h at 30 °C in a vacuum centrifuge (Speedvac, Eppendorf, Hamburg, Gemany). Dried organic phase aliquots were then re-suspended in 50 μL of MeOH/H_2_O (7:3, *v*/*v*) for LC-MS/MS analysis.

### 2.4. Lipid Quantification by LC-MS/MS

Target tandem mass spectrometry analysis was carried out in an Ultra-High-Performance Liquid Chromatography (UHPLC) system (Nexera X2, Shimadzu, Kyoto, Japan) coupled to a TripleTOF 5600+ mass spectrometer (Sciex, Foster City, CA, USA) equipped with a Turbo-V IonSpray and a Calibrant Delivery System (CDS). The calibration solution used was APCI Positive Calibration Solution (Sciex, Foster City, CA, USA) and the auto-calibration was performed after each of the five sample injections. For chromatographic separation, we used an Ascentis C18 column (15 cm × 2.1 mm, 2.7 μm; Supelco, Bellefonte, PA, USA), employing a method described by Godzien et al., 2015 [30] with some modifications as reported below. For each analysis, 10 μL of sample was injected. Elution was performed using H_2_O with 0.1% (*v*/*v*) of formic acid (phase A) and methanol with 0.1% (*v*/*v*) of formic acid (phase B), and the gradient conditions were as follows: 0 to 1.0 min., 30% B; 1.0 to 3.5 min., 80% B; 3.5 to 12.0 min., 100% B. The column oven temperature was maintained at 40 °C, the flow rate was 0.5 mL/min and the samples were at 4 °C in the autosampler. Collision energy (CE) and declustering potential (DP) parameters were determined for each compound to define the high-resolution multiple reaction monitoring (MRM^HR^) experiment. Compounds were fragmented via CAD using nitrogen as the collision gas. The parameters used for the mass spectrometer were: nebulizer gas (GS1), 50 psi; turbo-gas (GS2), 50 psi; curtain gas (CUR), 25 psi; electrospray voltage (ISVF), 4.5 kV; and turbo ion spray source temperature, 500 °C; dwell time, 100 ms, mass resolution < 1 ppm. The mass range of the product ion experiments was from 50 to 800 *m/z*. Data acquisitions were performed using Analyst™ software (Sciex, Foster City, CA, USA). Data obtained in the analysis was processed by using PeakView™ 2.1 and MultiQuant™ 3.0.2 software (Sciex, Foster City, CA, USA) for multiple fragment ion extraction, generation of the MRM^HR^ channels and for the quantitative analysis using a standard curve (1.75–500 ng/mL). The target MRM^HR^ parameters used for PAF-C16:0 (524.3678/184.0731), PAF-C18:0 (552.4006/184.0729), PAF-C18:1 (550.3859/184.0729), Lyso-PAF-C16:0 (482.3650/184.0740), Lyso-PAF-C18:0 (510.3905/184.0738), AEA (348.2897/287.2376), 2-AG (379.2841/287.2375)—(Parent ion/Precursor ion; *m*/*z*).

### 2.5. Protein Mediator’s Measurements

The levels of cytokines IL-6 and IL-10 were measured using a Cytometric Bead Array kit according to the manufacturer’s specifications (BD^TM^ Human Inflammatory Cytokine CBA Kit, Catalog No. 551811, Lot: 9341655, San Jose, CA, USA). The detection range of each cytokine was 5–5000 pg/mL. Data were acquired using a FACSCanto II flow cytometer and FACSDiva software (BD Biosciences, Franklin Lakes, NJ, USA). The data are presented as the mean fluorescence intensity for each serum cytokine. The levels of systemic sTREM-1 were measured in plasma using an ELISA kit (DuoSet-Human TREM-1 R&D System, Minneapolis, MN, USA) according to the manufacturer’s specifications.

### 2.6. RNA Extraction and Quantification

Total RNA was extracted from a 500 μL buffy coat using 0.5 mL of TRIzol reagent (Thermo Fisher Scientific, Carlsbad, CA, USA) and 0.2 mL of CHCl_3_. The upper phase containing RNA was transferred to a fresh tube with an equal volume of ethanol (1:1 *v*/*v*). After this, 0.7 mL of each sample was transferred to a spin cartridge containing a clear silica membrane to complete the extraction with on-column DNase treatment (Thermo Fisher Scientific, PureLink DNase Set, Carlsbad, CA, USA) per the manufacturer’s instructions (Thermo Fisher Scientific, PureLink RNA Mini Kit, Carlsbad, CA, USA). Quantity of RNA was determined using a Qubit Fluorometer (Thermo Fisher Scientific, Qubit RNA BR Assay Kit, Eugene, OR, USA), purity was determined using absorbance measurement ratios in a Nanodrop spectrophotometer (Thermo Fisher Scientific, Waltham, MA, USA) and quality of purified total RNA was determined using the RNA Integrity Number (RIN) values on a Bioanalyzer instrument (Agilent Technologies, Agilent 2100 Bioanalyzer system with RNA 6000 Nano kit, Waldbronn, Germany).

### 2.7. Data Transcriptome

Using the Clariom S Human Assay (Applied Biosystems^TM^, Clariom^TM^ S Assay human, Singapore) for single-sample (cartridge array) processing on the GeneChip 3000 instrument system (Applied Biosystems, GeneChip WT Pico Reagent Kit, Vilnius, Lithuania) in a high-throughput facility (Thermo Fisher Scientific, Microarray Research Services Laboratory, Santa Clara, CA, USA), whole transcript expression arrays were generated from 66 samples distributed in healthy controls (*n* = 12) and patients diagnosed with COVID-19 (*n* = 54).

### 2.8. Bioinformatics Analysis of Transcriptome Data

Target gene expression of enzymes involved in the eCBs and PAF pathways was obtained from whole blood leukocyte transcriptomic data available in the ArrayExpress database (http://www.ebi.ac.uk/arrayexpress) under accession number E-MTAB-11240 relative to the participants of this research consortium. This dataset contained pre-processed transcriptomic profiling from 66 samples distributed among healthy controls (*n* = 12) and patients diagnosed with COVID-19 (*n* = 54), subdivided into the different clinical classifications: Mild/Moderate (*n* = 26) and Severe/Critical (*n* = 28) and the use of glucocorticoids regardless of severity: COVID-19 non-CG (*n* = 19) and COVID-19 GC (*n* = 35). The bioinformatic analyses were performed using *R 4.1.2 libraries* [31] the RStudio environment [32] and Bioconductor libraries [33]. We used preprocessed expression data that had been deposited at the probe set level to obtain gene-based expression by collapsing the probes using the maximum mean method with the collapse Rows function of the *WGCNA 1.71* package and using gene-to-probe annotation that was available in the same dataset. Differential expression for the whole transcriptome between previously described clinical groups was performed with *limma 3.50.1*. Differentially expressed genes (DEGs) were defined in at least one pair of clinical groups using Benjamini and Hockberg’s [34] adjusted *p*-values of 0.05. Graphical representations of generated data were constructed using *ggplot2 3.3.5* and *heatmap 1.0.12*.

### 2.9. Statistical Analysis

The data were evaluated for a normal distribution using the Shapiro–Wilk normality test and D’Agostino and Pearson test. The parametric data were analyzed using unpaired *t*-tests or one-way ANOVA followed by Tukey’s multiple comparison tests. For non-parametric data, Mann–Whitney or Kruskal–Wallis test were used, followed by Dunn’s post-tests. The *chi*-square test was used to assess associations among categorical variables and COVID-19. The results were obtained using GraphPad Prism^TM^ software (version 9.0, Boston, MA, USA) and the differences were considered statistically significant at *p* < 0.05. The dependence on multiple variables were analyzed using significant Spearman’s correlations at *p* < 0.05. Data were presented in the Correlation matrix using the R package *qgraph* [35]. The log^2^ of normalized gene expression profiles for analyzed groups is shown as boxplots. Significant differences in transcript expression correspond to Benjamini and Hockberg adjusted *p*-values obtained from whole transcriptome differential expression analysis considering a threshold of <0.05 in at least one pair of clinical groups.

## 3. Results

### 3.1. Participants Demographic Data and Clinical Characteristics

In detail, the clinical features of the subjects who signed up for the study are described in Appendix A. Parameters linked with age (*p* < 0.0001), BMI (*p* = 0.0015) and comorbidities such as hypertension (*p* = 0.0041) and diabetes (*p* = 0.0228) were significantly different between COVID-19 patients and the control group. Additionally, the patients had persistent symptoms such as dyspnea (57%), fever (31.5%) and myalgia (23.5%). In the COVID-19 group, the laboratory data revealed changes in total leukocyte counts (*p* = 0.0295), followed by significant neutrophilia (*p* < 0.0001) and lymphopenia (*p* = 0.0001). Regarding the hospitalized patients, 45% were in the infirmary and 28.5% were in the intensive care unit. The pharmacological treatment includes GCs for 61.5%, azithromycin for 58.5%, ceftriaxone for 46.5%, oseltamivir for 32.5%, hydroxychloroquine for 15%, colchicine for 2.5% and ivermectin for 4% of total patients, consistent with the care procedures in place at the time of hospitalization.

### 3.2. Production of AEA and 2-AG Is Elevated in Severe COVID-19 Patients with a Link to GCs-Treatment

Plasma levels of the eCBs, AEA and 2-AG were determined by LC-MS/MS. According to the severity of the condition, we quantified the synthesis of these lipid mediators. Analyses of descriptive data revealed an increase in AEA (Figure 1a) and 2-AG (Figure 1b) production in severe/critical COVID-19 patients versus controls and mild/moderate patients. Therefore, we separated the groups based on the severity and utilization of pharmaceutical therapies for GCs. While GCs had no influence on AEA levels (Figure 1c), they did enhance 2-AG production (Figure 1d) in the comparison of mild/moderate (non-GC versus GC, *p* = 0.0120) and severe/critical (non-GC versus GC, *p* < 0.0001) patients, suggesting that GCs may have an effect on the modulation of the 2-AG pathway.

### 3.3. Diminished Levels of Lyso-PAF and PAF, and Different Ratios in Lyso-PC and Lyso-PE Species in Severe COVID-19 Patients with the Use of GCs

When we compared PAF plasma production in different COVID-19 severities, we discovered an inverse phenomenon to what happens with eCB production. Descriptive analyses showed a reduction of Lyso-PAF (16:0) (Figure 2a), Lyso-PAF (18:0) (Appendix A), PAF (16:0) (Figure 2b) and PAF (18:0) (Appendix A) production in severe/critical COVID-19 patients compared to mild/moderate patients (*p* < 0.0001) and control subjects (*p* < 0.0001). In addition, when dividing the groups according to pharmacological treatments with GCs, we observed that the individuals who used GC showed a reduction in the levels of Lyso-PAF (16:0) (Figure 2c) and PAF (16:0) (Figure 2d), suggesting a possible influence of GCs on the production of these mediators.

In addition to quantifying the production of these lipid mediators, we investigated other Lyso-PC and Lyso-PE species using semi-quantitative analyses by area ratio average to see if the use of GCs could affect membrane lipid modulation and serve as a source for PAF and eCB initiation metabolism. We observed that the Lyso-PC (16:0) level is increased in patients in the severe/critical COVID-19 group (approximately 70%) compared to the other groups (Figure 2e). However, the severe/critical patients who received GC-treatment showed a reduction in Lyso-PC (16:0) levels (Figure 2g). The other Lyso-PC species, (20:4), (18:2) and (18:0), maintain the production reasonably constant in all groups. We also demonstrated an increased area ratio of Lyso-PE (16:0) and a decreased of Lyso-PE (18:2) in the plasma of severe/critical COVID-19 patients, compared to control and mild/moderate patients (Figure 2f). In the severe/critical group, GC treatment partially reversed the production of Lyso-PE (18:2) (Figure 2h). The other Lyso-PE lipids, (22:5), (20:4), (18:1) and (18:0), did not show vast differences between the patient groups.

### 3.4. Modified Genetic Expression Profile of Metabolic Enzymes Implicated in the eCBs and PAF Pathway in COVID-19 Patients Treated with GCs

We compared the blood cells gene expression regulation of enzymes involved in the PAF and eCBs pathways for different severity in COVID-19 patients using a heatmap (Appendix A) and PCA analysis (Appendix A). We noted that the mild/moderate patients tended to cluster apart from the severe/critical and control groups; however, we did not identify any significant differences between these patient groups. In a similar manner, we made the decision to conduct an analysis of the gene expression based on the treatment with GCs, independent of the severity of COVID-19, in the groups COVID-19 (GC), COVID-19 (non-GC) and control. The membrane Alkyl-20:4 GPE is the source and starting point for the de novo and remodeling PAF pathways, which both require the production of Lyso-PAF as an intermediary. Therefore, membrane lipid hydrolysis by phospholipase A2 (PLA2) is necessary for the formation of lyso-PAF or PAF. The gene expression of PLA2 isoforms, PLA2G4A and PLA2G5, were upregulated, while PLA2G6 and PLA2G7 were downregulated in COVID-19 patients using GCs, compared to non-treated COVID-19 patients and controls (Figure 3a). There were no significant differences between groups in the expression of PLA2G3 and PLA2G2A (Appendix A). Lyso-PAF is acetylated by acetyl-CoA and Lyso-PAF acetyltransferase (LPCAT). Gene expression of these enzyme isoforms is elevated for LPCAT2 in GCs-using patients, while no alterations were identified for LPCAT1 expression (Figure 3b). PAF degradation requires the removal of the acetyl group, which is conducted by PAF-acetylhydrolase (PAFAH). We showed a significant increase in the gene expression of PAFAH1B1 in the COVID-19 GCs group (Figure 3c), but there was no change in the expression of PAFAH1B2 (Appendix A) and PAFAH1B3 (Appendix A). Surprisingly, GCs therapy elevated PAF-receptor (PTAFR) gene expression relative to untreated COVID-19 patients, which may be associated with pro-inflammatory effects. However, these expressions were below the control group’s baseline level (Figure 3d).

In the initial stages of AEA synthesis, phosphatidylethanolamine (PE) is converted into N-arachidonoyl phosphatidylethanolamine (NAPE). Considering the production of AEA in COVID-19 patients treated with GCs, we observed that the gene expression of the enzyme NAPE-specific phospholipase D (NAPE-PLD), which is responsible for hydrolyzing NAPE to form AEA, was downregulated (Figure 3e), whereas there was no effect on the expression of the FAAH gene, which is involved in the catabolism of AEA to arachidonic acid (AA) (Figure 3f). On the other hand, the phospholipid-inositol is transformed into diacylglycerol (DAG) via the action of phospholipase C (PLC) during the process of the metabolism of 2-AG. The expression of PLCβ1 and PLCβ2 was increased and downregulated, respectively, in GCs-treated COVID-19 patients (Figure 3g). GCs treatment had no effect on the genes PLCβ3 (Figure 3h) and PLCβ4 (Figure 3i). DAG can be processed sequentially by diacylglycerol lipase (DAGL) to generate 2-AG, but we did not notice any variations in DAGLB expression across patient groups (Figure 3h). Nonetheless, we detected an increase in the gene expression of MGLL, which is orthologous to monoacylglycerol lipase (MAGL) (Figure 3i) in COVID-19 patients receiving GCs, indicating higher 2-AG breakdown by MAGL to glycerol and AA synthesis. Lastly, gene expression of CB2 cannabinoid receptors CNR2 (Figure 3d) was raised in GCs-treated COVID-19 individuals compared to control, and there was a trend toward increased expression of CNR1 (Appendix A). Detailed gene fold change values and *p*-values obtained from the differential expression analysis between the analyzed groups are included in Appendix A.

### 3.5. Plasma 2-AG and PAF Levels Correlated with Inflammatory Markers of COVID-19

We performed a correlation matrix using the Spearman’s test to investigate the interaction between systemic inflammatory parameters and the levels of 2-AG and PAF C16 (16:0) (Figure 4a). Increased 2-AG production correlated moderately and positively with the anti-inflammatory cytokine IL-10 (*p* = 0.0036) and with absolute levels of circulating lymphocytes (*p* = 0.0195), but moderately and negatively with IL-6 (a pro-inflammatory cytokine) (*p* = 0.0398), absolute levels of circulating neutrophils (*p* = 0.0380) and sTREM-1 (*p* = 0.019). This indicates that 2-AG has a potential anti-inflammatory impact, aiding in the regulation of COVID-19 host immunological response. However, PAF (16:0) production was only moderate and positively linked with the absolute values of circulating lymphocytes (*p* = 0.0013); details and *r* values are included in Appendix A. Subsequently, we categorized absolute neutrophil counts (Figure 4b), lymphocytes (Figure 4C), platelets (Appendix A), coagulation markers and the international normalized ratio (INR) (Appendix A) based on the GCs therapy and severity of COVID-19. As expected, severe/critical patients had higher neutrophil counts (Figure 4b) and lower lymphocyte counts (Figure 4c) than controls and mild/moderate patients. However, only the mild/moderate group demonstrated a significant difference between GCs-treated and untreated individuals, with neutrophil counts increasing and lymphocyte counts decreasing in GCs-treated patients. Then, we hypothesized that this effect was not a result of the GCs treatment, but rather of the increasing severity noticed in some moderate patients, which was the reason to begin using GCs. In addition to the inflammatory indicators, we observed no significant changes in the thrombotic markers in these patients (Appendix A).

### 3.6. The Use of GCs Alters the Metabolism and Production of Lipid Mediators during COVID-19

In order to understand whether the plasma production of eCBs, PAF and the Lyso-PC and Lyso-PE families are correlated, we used the Spearman’s test to investigate the interaction between these lipids species in COVID-19 (Figure 5a). 2-AG levels show moderate and positively correlation with Lyso-PC 22:5 (*r* = 0.352), Lyso-PC 16:0 (*r* = 0.378), Lyso-PC 18:2 (*r* = 0.225) and Lyso-PC 18:0 (*r* = 0.221) with significant *p* values (*). Additionally, we observed that PAF C16 is positively correlated with all Lyso-PC species: Lyso-PC 22:5 (*r* = 0.430), Lyso-PC 20:4 (*r* = 0.223), Lyso-PC 18:2 (*r* = 0.364), Lyso-PC 16:0 (*r* = 0.510) and Lyso-PC 18:0 (*r* = 0.369) and three lipids from the Lyso-PE family, Lyso-PE 20:4 (*r* = 0.308), Lyso-PE 18:0 (*r* = 0.190) and Lyso-PE 18:2 (*r* = 0.360) with significant *p* values (*) (the details are described in Appendix A). Plasma lipid analysis, performed using target LC-MS/MS, demonstrated the enhanced release of AA in COVID-19 patients treated with GCs compared to the non-GCs group (*p* = 0.0280) (Figure 5b), these data supporting the evidence of an active metabolism to produced lysophosphatidic lipids species.

Schematic illustration of our model in COVID-19 with the following: (i) the 2-AG pathway, which could create AA as a secondary product, is launched by the membrane glycerophospholipids being cleaved by PLC, creating diacylglycerol (DAG), which then undergoes the action of diacylglycerol lipase (DAGL), forming 2-AG, which can be metabolized by the action of MAGL to form AA and glycerol; (ii) the synthesis of AEA is initiated by the action of PLA2, which generates N-arachidonoyl phosphatidyl ethanol (NAPE), which is then metabolized by NAPE phospholipase D (NAPE-PLD), resulting in the formation of AEA, which might be metabolized by FAAH to produce AA; (iii) the production of PAF, which originates from the remodeling pathway via PLA2-mediated synthesis, cleaving membrane glycerophospholipids, generating its direct precursor Lyso-PAF and releasing AA. An acetyltransferase (LPCAT) acetylated Lyso-PAF to produce PAF. The GC treatment had the effect in COVID-19 of increasing production of 2-AG and AA while decreasing formation of PAF (Figure 5c).

## 4. Discussion

Inflammation lies at the heart of aggravated infectious conditions, such as the severity of COVID-19 infections; the deadly outcomes of this disease are systemic hyperinflammation and multi-organ failure [36]. Until the introduction of a vaccine, pharmaceutical therapy remains a crucial method for combating the current pandemic. Due to their anti-inflammatory and immunosuppressive properties, GCs have garnered significant attention. Few studies, however, have examined the impact of lipid mediators, such as the eCB system and PAF group, to these effects. In addition, PUFAs are produced from phospholipids, specifically blood cells, during the early phase of the host’s reaction to pathogens [37]. In this investigation, we hypothesized that immune cells engaged in the COVID-19 response could release eCB and PAF, hence mediating systemic symptoms. In fact, patients with severe or life-threatening SARS-CoV-2 infection show higher plasma levels of eCB species (AEA and 2-AG). In contrast, the generation of Lyso-PAF and PAF species was reduced in plasma from patients with severe or critical diseases. In order to improve the efficacy of treatment for COVID-19 patients, medication assisted by GCs was advised for patients in our cohort, particularly those with severe COVID-19 symptoms. Interestingly, these effects of higher plasma levels of 2-AG and lower PAF were most pronounced in GCs-treated patients. This information makes it possible to identify novel molecular pathways for the creation of targeted medicines.

Lipid mediators play a crucial function as intercellular signaling molecules; hence, viruses manipulate lipid signaling and metabolism to facilitate viral replication [38]. As a result of the enzymatic metabolism of phospholipids, eCB-related metabolites are produced. The AA-based compounds AEA [39] and 2-AG [40] are produced on demand from lipid precursors within plasma membranes in response to stimuli such as intracellular calcium increases [41]. Beyond their role as modulators of the central nervous system (CNS), eCBs are immunological mediators with anti-inflammatory and antitumor activity [42,43,44], depending on receptor binding (CB1 or CB2). CB1 receptors are widely expressed in the CNS, but have also been detected at lesser levels in peripheral tissues, such as the lung [45]. CB2, the second eCB receptor, was discovered in immune cells [46] and is best known for its immunosuppressive effect [47]. In certain immune cells, the activation of CB2 receptors by the binding of 2-AG and AEA [48,49] might decrease cytokine production, diminish the recruitment of neutrophils and M1 macrophages and ultimately result in decreased levels of reactive oxygen species (ROS) [50]. In our investigation, patients with severe/critical COVID-19 had considerably greater plasma concentrations of AEA and 2-AG than healthy controls, with the latter being 20-fold higher than the former. AEA is a low-effectiveness, high-affinity agonist of the CB1 receptor. In contrast, 2-AG has a lesser affinity for CB1 receptors but greater effectiveness. Interestingly, 2-AG is much more common than AEA in the CNS. This is partly because 2-AG acts as an inert intermediate in multiple lipid metabolic pathways [51], especially as a bulk carrier of AA, which can be quickly moved to help make prostaglandins [52]. It is known that AEA can reduce ARDS through the activation of anti-inflammatory pathways in regulatory T cells [53]. Moreover, therapy with AEA lowered the expression of IL-6 in an ARDS experimental model [54]. In another study, it was shown that AEA and 2-AG reduce pro-inflammatory cytokines and increase anti-inflammatory cytokines in individuals with HIV [55]. Indeed, we observed a positive association between 2-AG production and IL-10 and lymphocyte counts and a negative correlation with different COVID-19 markers of hyperinflammation, such as IL-6 [56], sTREM-1 [57] and neutrophil counts [56]. Although high levels of 2-AG did not prevent the severe course of COVID-19, they did demonstrate anti-inflammatory capabilities.

Another key lipid mediator identified in this study was PAF, which acts as an endogenous agonist at platelet activating factor receptor (PAFr) sites expressed on the surface of leukocytes, endothelial cells, platelets and a variety of other cell types, boosting leukocyte chemotaxis and acting as a potent mediator of inflammation, particularly in response to microbial or viral infectious processes [58,59]. Regarding COVID-19, PAF is a highly pyrogenic substance [60] and it influences the activity of angiotensin converting enzyme 2 (ACE2) [61], which is a receptor utilized to facilitate the entry of SARS-CoV-2 into cells [62]. Moreover, PAF is a strong thrombosis mediator [8,63]. In addition to the theorized relationship between PAF functions and COVID-19 pathophysiology, we demonstrated for the first time the formation of Lyso-PAF and PAF species in plasma of COVID-19 patient. Curiously, the levels of PAF seem to be higher in mild/moderate COVID-19 patients compared to healthy controls, while the production decreases in individuals with severe/critical disease. Additionally, several studies are looking into the efficacy of PAF-inhibiting pharmaceutical drugs in treating COVID-19 [7,64].

The correlation between the severity and prognosis of COVID-19 and the level of inflammatory response is high. SARS-CoV-2 infection induces inflammation from the outset. When the immune system is unable to effectively eliminate the virus, pulmonary and systemic hyperinflammation ensues [65]. A balanced immune response to SARS-CoV-2 contributes to the virus’s effective eradication in the majority of patients. Nevertheless, an inadequate immune response causes some COVID-19 patients to have a severe dysregulated inflammatory process and a cytokine storm [65], which is defined by the activation of the innate immune system due to poor viral clearance, inadequate levels of type I interferons (IFN-I), enhanced neutrophil extracellular traps (NETS) and other miscellaneous mechanisms [66,67]. GCs are a globally accessible, low-cost anti-inflammatory treatment that is available to everyone. Due to their immunosuppressive efficacy, GCs are among the medications that are still being studied for the treatment of COVID-19. In addition, GCs are commonly employed in the treatment of disorders closely related to COVID-19, such as severe acute respiratory syndrome (SARS) and Middle Eastern respiratory syndrome (MERS) [68,69,70]. The most persuasive evidence regarding the use of GCs in COVID-19 patients comes from the RECOVERY trial, in which a range of possible treatments was compared in patients who were hospitalized with COVID-19, in randomly assigned patients that received oral or intravenous dexamethasone [28]. This study indicated that GCs therapy was advantageous for patients with severe COVID-19 compared to a control group, who were being treated more than 7 days after symptom onset, when inflammatory lung damage is likely to have been more common, but had no meaningful effect on patients who did not require oxygen mechanical breathing assistance [28]. Indeed, we discovered that almost 60% of COVID-19 patients in our cohort received GCs medication for at least 3 days before to enrollment in this study, and we investigated the influence of the GCs treatment on eCB and PAF pathway modification during COVID-19.

Non-genomic and genomic signaling systems are involved in the anti-inflammatory effects of GCs [71,72]. GCs decrease T-cell activation and the generation of pro-inflammatory cytokines at a non-genomic level [73]. However, the bulk of anti-inflammatory mechanisms are linked to direct modulation of gene transcription, such as the regulation of genes that control cell activation and the production of inflammatory mediators [23], as well as potentials through suppression of NF-kB activities [74,75]. GCs limit gene expression of pro-inflammatory factors and enzymes, such as cytosolic PLA2, which controls the release of AA from phosphatidylcholine (PC) and phosphatidylethanolamine (PE) [76]. By destabilizing COX-2 mRNA, dexamethasone suppresses COX-2 synthesis and shifts AA metabolism away from the generation of pro-inflammatory signaling molecules [77]. We found that GC treatment was substantially connected with the effects of enhanced 2-AG release and decreased PAF formation in plasma after dividing COVID-19 patients into groups treated with GCs and those not treated with GCs. However, there was no effect of GCs therapy on neutrophils, lymphocytes or platelet counts in the blood of severe/critical COVID-19 patients.

Despite considerable differences in receptor selectivity between AEA and 2-AG, both eCBs are generated in response to an increase in intracellular Ca^2+^ [78,79]. Briefly, AEA is produced from N-acyl-phosphatidylethanolamine (NAPE) by NAPE-specific phospholipase D (NAPE-PLD) in a single step or by other pathways comprising two or more catalytic stages [80,81]. Alternatively, 2-AG is derived from diacylglycerol (DAG) via DAG lipase (DAGL) [82]. The rate-limiting and Ca^2+^-sensitive phase in AEA and 2-AG production is the generation of NAPE and DAG from PE by N-acyltransferase/cPLA2 and phosphoinositides (PI)/PC by phospholipase C (PLC) [82]. The final fundamental components of eCB signaling are the enzymes responsible for AEA and 2-AG breakdown and, eventually, their half-life [83]. AEA is degraded by fatty acid amide hydrolase (FAAH) into free AA and ethanolamine, whereas 2-AG is predominantly hydrolyzed by monoacylglycerol lipase (MAGL) and to a lesser extent by ABHD6 and ABHD12 [84,85] into AA and glycerol [82,84]. Both AEA and 2-AG oxidation may involve COX-2 [86]. In fact, we observed that GC therapy elevated cPLA2 enzyme-corresponding genes (PLA2G4A and PLA2G5) in COVID-19 patient blood cells. However, the expression of genes encoding enzymes in the AEA pathway, such as NAPE-PLD, was downregulated in GCs-treated patients, indicating that GCs was not responsible for the increased synthesis of AEA in severe/critical COVID-19 patients. In contrast, the expression of PLCB1 (the PLC enzyme) was elevated in the GCs-treated group, which corresponds to the rate-limiting phase of DAG synthesis in the 2-AG metabolism pathway. In addition, the MGLL gene expression (MAGL enzyme) was elevated in the same samples, indicating an additional mechanism for the release of AA by the GCs-treatment effect. In addition, we hypothesized that GCs may positively control Ca^2+^ release as a secondary signal and boost 2-AG formation in plasma.

The chemical structure of PAF was reported as 1-Oalkyl-2-acetyl-sn-glycero-3-phosphocholine [87], and it is synthesized in the cells via two pathways: (i) the de novo metabolic pathway; and (ii) the cytosolic phospholipase A2 (cPLA2)-dependent remodeling pathway [88]. The first process consists of three enzymatic reactions to create PAF from its precursor, 1-alkyl-2-lyso-sn-glycero-3-phosphate [89], which is acetylated by an acetyltransferase, dephosphorylated by a phosphohydrolase and finally combined with choline [59]. In the second route, calcium-dependent PLA2 and a coenzyme A-independent transacylase convert the immediate precursor of PAF (acyl-PAF) into Lyso-PAF [90]. Lyso-PAF is converted to PAF by Lyso-PAF acetyltransferase (LPCAT) [91,92]. In contrast, the acetyl hydrolase enzyme (PAFAH) converts PAF activities into lyso-PAF, which has no biological activity [88]. Recent research demonstrated that PAF formation is a continuous process that does not need activation of non-inflammatory cells by the membrane remodeling mechanism [93], which could explain the elevated PAF levels in our control plasma samples. Concerning the PAF-remodeling pathway, we discovered that GCs therapy decreased the expression of the genes encoding calcium-independent PLA2 (PLA2G6) and lipoprotein-associated PLA2 (PLA2G7), which both act in LDL, the principal source of Lyso-PAF in plasma [94]. In addition, the same group of patients produced less Lyso-PC (16:0), highlighting the function of GCs in the control of PLA2G7. Indeed, we observed elevated LPCAT2 and PAFAH1B1 gene expression in GCs-treated patient cells. The observations indicated that both enzymes involved in the metabolic creation of PAF from Lyso-PAF and the catabolism of PAF to Lyso-PAF were significantly expressed, and the finding of decreased production of PAF in plasma may be the consequence of a post-translational regulation mechanism induced by GCs. Since PAF could activate physiological pathways that contribute to the pathogenic effects of COVID-19, it is possible that pharmacological agents that directly inhibit the activity of PAF or modulate the cycle of PAF synthesis and degradation could reduce morbidity and mortality caused by COVID-19 [95,96,97].

Our study distinguishes itself from others since it comprises a bigger group of individuals who have been clinically described in depth. Nonetheless, it is subject to a number of limitations. Because persons who are at risk for severe COVID-19 may already be suffering from dysregulation of lipid metabolism due to underlying health issues [98,99], it is vital to keep this in mind when evaluating any metabolic perturbations, and especially those of lipid mediators. We have not taken into account the chronic conditions and secondary infections that may also contribute to lipid metabolic dysregulation. Our samples are only representative of the community in their respective regions; hence, bigger cohort studies are required to establish the predictive effect in key persons infected with COVID-19. This may be medically complex due to the continual changes in patient management and treatment approaches, including vaccination, and the emerging SARS-CoV-2 variations. A lack of uniformity clinical testing utilized by the hospitals participating in this study, for the care of patients with COVID-19, hindered the accuracy of additional laboratory data. The absence of a suitable control group with GCs-therapy is another drawback of our study, in addition to an imbalanced cohort throughout disease stages and the approach of pharmacological treatment. Since circulating signaling lipids might fluctuate during the day, we collect blood early in the morning after an overnight fast [100]. For lipidomics analysis, blood plasma is better than serum because it prevents lysophospholipids and other compounds from having a skewed profile, and a small amount of methanol is added to samples to keep them from oxidation degradation. The beneficial effect of GCs in severe viral respiratory infections is likely to depend on the timing of administration, dosage and patient type [101].

## 5. Conclusions

In the context of severe infections, GCs have a convoluted history. A clear indication for GCs in severe COVID-19 has been established recently [28]. This may be a precursor to the increased usage of GCs in critical illnesses. Understanding the heterogeneity of the operationalization of GCs within the precision medicine framework can be a critical prerequisite for this effort. Furthermore, this study provided insight into a new pharmacological mechanism of GCs in the control of the inflammatory response during COVID-19. The lipid mediators 2-AG (an eCB metabolite) and PAF were observed to be elevated and lowered, respectively, in the plasma of COVID-19 patients, independent of illness severity symptoms but in relation to GCs treatment. With these results, we might hypothesize several strategies for the therapy of COVID-19 disease by inhibiting or boosting diverse phases of the metabolism of these lipid mediators.

## Figures and Tables

**Figure 1 viruses-15-00573-f001:**
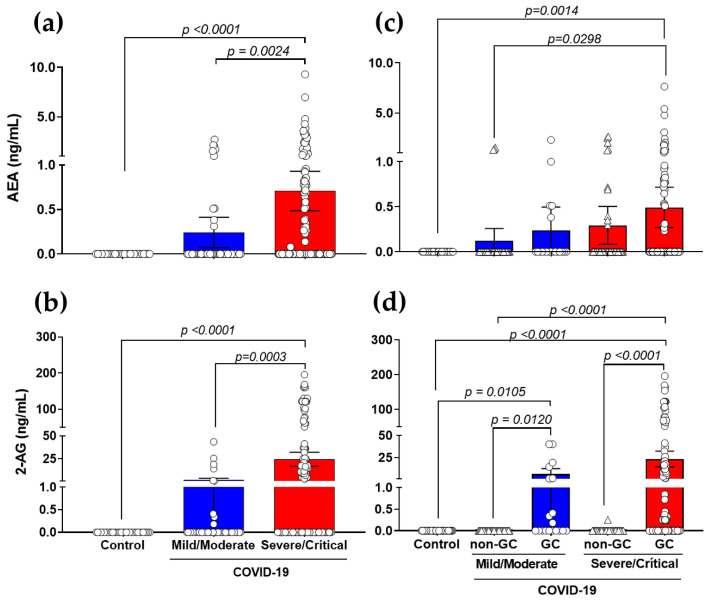
Production of plasma AEA and 2-AG in patients with COVID-19 using GC treatment. Levels of AEA (**a**) and 2-AG (**b**) in COVID-19 patients (mild/moderate, *n* = 55 and severe/critical, *n* = 145) compared to healthy controls (*n* = 35). COVID-19 patients who use/do not use GC were segregated into mild/moderate (non-GC, *n* = 35 vs. GC, *n* = 20) and severe/critical (non-GC, *n* = 42 vs. GC, *n* = 103), showed production of AEA (**c**) and 2-AG (**d**) compared to healthy controls. Statistical analyses were performed using the Kruskal–Wallis multiple comparison test (non-parametric), followed by Dunn’s post-test. Data are expressed as median in boxplot graphs with minimum and maximum values with a confidence interval of +/− 95%. Significance levels shown are based on statistically significant *p*-values between groups, considering significant with *p* < 0.05.

**Figure 2 viruses-15-00573-f002:**
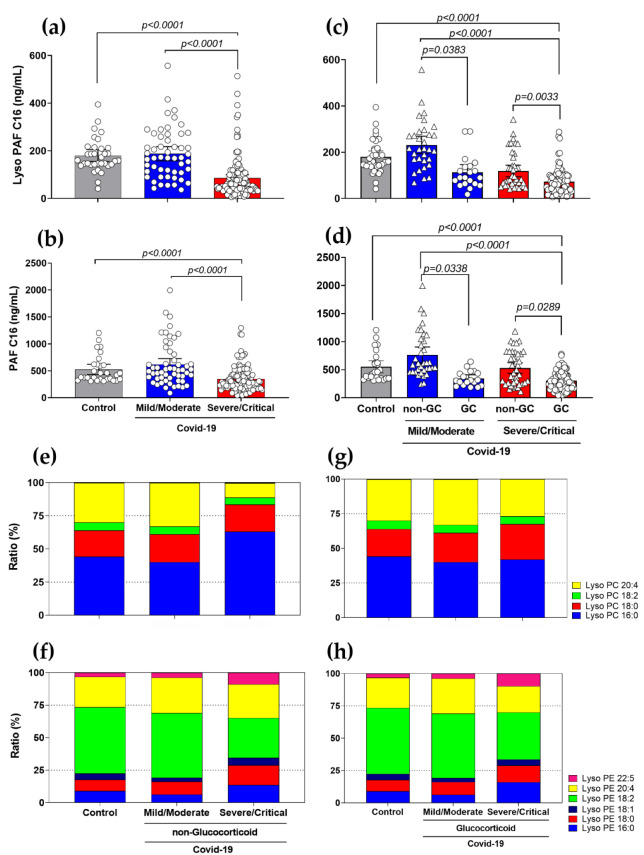
Production of Lyso-PAF C16, PAF C16, and relative abundance of Lyso-PC and Lyso-PE species in COVID-19 patients with the use of GCs. Levels of Lyso-PAF (16:0) (**a**) and PAF (16:0) (**b**) in COVID-19 patients (mild/moderate (*n* = 55) and severe/critical (*n* = 145) compared to healthy controls (*n* = 35). COVID-19 patients who use/do not use GCs were segregated into mild/moderate (non-GC, *n* = 35 vs. GC, *n* = 20) and severe/critical (non-GC, *n* = 42 vs. GC, *n* = 103) show significant differences in Lyso-PAF (16:0) (**c**) and PAF (16:0) (**d**) compared to healthy controls. Relative abundance (ratio %) production of Lyso-PCs (**e**) and Lyso-PEs (**f**) in individuals not treated with GCs (mild/moderate, *n* = 35 and severe/critical, *n* = 42); Lyso-PCs (**g**) and Lyso-PEs (**h**) of GC-treated individuals (mild/moderate, *n* = 20 and severe/critical, *n* = 103). Statistical analyses were performed using the Kruskal–Wallis multiple comparison test (non-parametric), followed by Dunn’s post-test. Data are expressed as median in boxplot graphs with minimum and maximum values with a confidence interval of +/−95%. Significance levels shown are based on statistically significant *p*-values between groups, significant for *p* < 0.05. Area ratio: area ratio between the metabolite and the correspondence internal standard.

**Figure 3 viruses-15-00573-f003:**
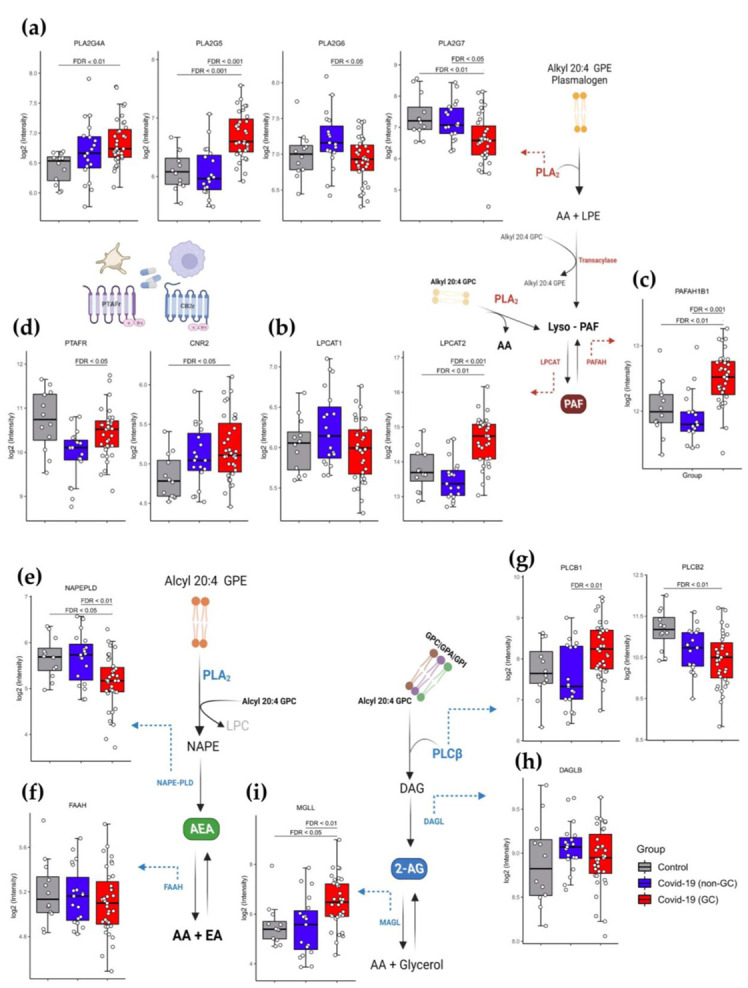
The influence of GCs in the gene expression of enzymes and receptors related to the eCB and PAF pathway in whole blood leukocyte transcriptomic data. Schematic representation of the PAF formation pathway, involved the genes: (**a**) PLA2G4A, PLA2G5, PL2G6 and PL2G7; (**b**) LPCAT1 and LPCAT2; (**c**) PAFAH1B1 and (**d**) PTAFR and CNR2. On behalf of eCB pathway for AEA formation, the gene expression: (**e**) (NAPEPLD) and (**f**) (FAAH); and 2-AG formation: (**g**) PLCB1; PLCB2; (**h**) DAGLB; (**i**) MGLL. Differential expression was carried out between Healthy control (*n* = 12), COVID-19 non-CG (*n* = 19), and COVID-19 GC (*n* = 35) groups. The log2 of normalized gene expression profiles for analyzed groups were showed as boxplots. Significant differences in transcript expression was accessed using Benjamini and Hockberg adjusted *p*-values to controlling false discovery rate (FDR) obtained from whole transcriptome differential expression analysis considering a threshold of FDR < 0.05. Appendix A contained gene fold change values, nominal and FDR adjusted *p*-values obtained from the differential expression analysis between the analyzed groups.

**Figure 4 viruses-15-00573-f004:**
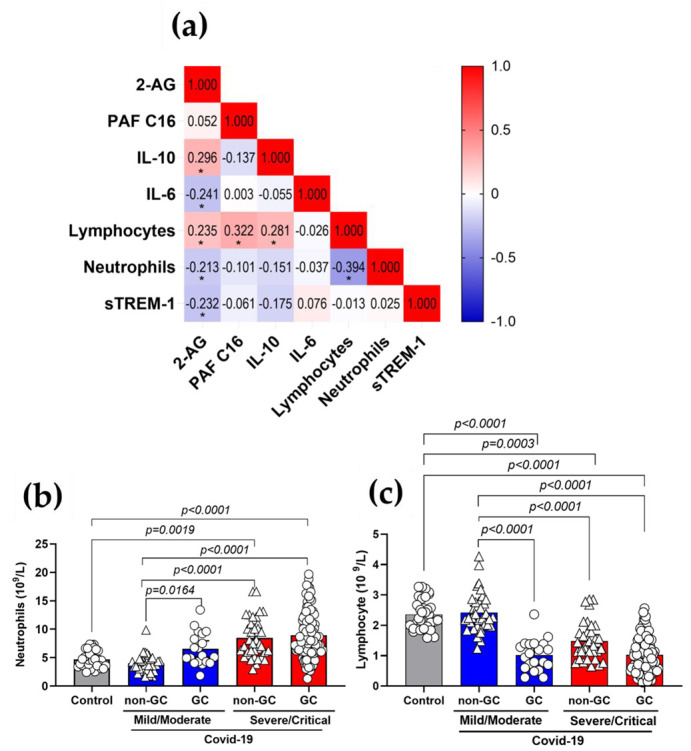
Correlations of 2-AG and PAF C16 with inflammatory markers of COVID-19 patients. (**a**) Correlation matrix demonstrating interactions between levels of 2-AG, PAF (16:0) and the inflammatory parameters, IL-10, IL-6, sTREM-1, lymphocytes and neutrophil counts. Color scale sidebar indicates correlation coefficients (*r*), color-coded: red, positive correlation; blue, negative correlation; the intensity of the color represents the intensity of the correlation. Values adjust between −1.0 and 1.0. The significance levels indicated with gray asterisks are based on the *p* < 0.05 of the Spearman’s correlation coefficient (*r* ) *. (**b**) Absolute values of neutrophils and (**c**) lymphocytes in patients GC-treated or not with COVID-19 mild/moderate (non-GC, *n* = 35 vs. GC, *n* = 20) and severe/critical (non-GC, *n* = 42 vs. GC, *n* = 103). Statistical analyses were performed using the Kruskal–Wallis multiple comparison test (non-parametric), followed by Dunn’s post-test. Data are expressed as median in boxplot graphs with minimum and maximum values with a confidence interval of +/−95%. Significance levels shown are based on statistically significant *p* < 0.05 between groups.

**Figure 5 viruses-15-00573-f005:**
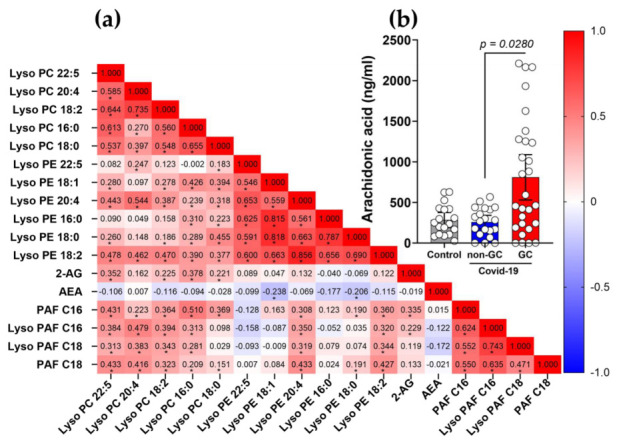
Altered profile of eCBs and PAF induced by the use of GCs in patients with COVID-19. (**a**) Correlation matrix between lipid mediators species in individuals using GCs. The color scale sidebar indicates the correlation coefficients (*r*), color-coded: red, positive correlation; blue, negative correlation; the intensity of the color represents the intensity of the correlation. Values adjust between −1.0 and 1.0. Significance levels indicated with gray asterisks are based on the *p*-value < 0.05 of the Spearman’s correlation coefficient (*r*)*. (**b**) Production of arachidonic acid (AA) in COVID-19 patients with the use of GCs, healthy controls (*n* = 18), non-GC (*n* = 22) and GC (*n* = 30). Statistical analyses were performed using the Kruskal–Wallis multiple comparison test (non-parametric), followed by Dunn’s post-test. Data are expressed as median in boxplot graphs with minimum and maximum values with a confidence interval of +/−95%. Significance levels shown are based on statistically significant *p* < 0.05 values between groups. (**c**) Schematic representation of the eCBs and PAF pathways in COVID-19 patients and the regulation of GCs in this model. (Created with BioRender.com, Agreement number: RJ24TV89SK).

## Data Availability

Normalized transcriptomic data from the participants’ whole blood leukocytes and associated metadata were deposited on the Array Express database at EMBL-EBI (www.ebi.ac.uk/arrayexpress, accessed 2 November 2022) under accession number E-MTAB-11240.

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
