# Peer review of "The Interplay among Glucocorticoid Therapy, Platelet-Activating Factor and Endocannabinoid Release Influences the Inflammatory Response to COVID-19"

_viruses, 2023, doi:10.3390/v15020573_

Round 1

Reviewer 1 Report

Commendable job to the authors for carrying out an extensive evaluation of the lipid mediators in COVID19 related inflammation.

Few changes that might help the readers understand the paper better.

1. Title does not represent the content of the paper. It is not a clinical application- You have conducted a cross-sectional bench side study.

2. Lines 74 to 75 (reference 11) need to expand the mechanism of immune suppression in sepsis and the role of lipid mediators in the same. Maybe in brief.

3. Lipid regulation of viral replication should be supported with better references in addition to 21. (lines 94-96)

4. Role of glucocorticoids in severe COVID-19 is not controversial now. It has been proven to reduce mortality. So we should modify the line 97-98

5. This is cross-sectional study (line 108).

6. How many subjects of COVID-19 who were diagnosed on the basis of serological assays were classified to have severe disease or acute disease?

7. Subjects on immunomodulators- were they included or excluded? In either case must be mentioned in the methods section.

8. Have the authors tried to evaluate the samples of same cases- before and after the adminstration of GC? 

9. Severity correlation with respect of pulmonary functions in the form of PaO2/FiO2 ratio can be added, if the authors have that data. This will have better correlation value as comapred to hematological parameters which might be affected due to medicaitons also in addition to the disease severity.

Author Response

Response to Reviewer 1 Comments:
Commendable job to the authors for carrying out an extensive evaluation of the lipid mediators in COVID19 related inflammation.
Few changes that might help the readers understand the paper better.

1. Title does not represent the content of the paper. It is not a clinical application- You have conducted a cross-sectional bench side study.
Response: We agree with the referee´s concern. Then, we modified the title: “The interplay among glucocorticoid therapy, platelet-activating factor an endocannabinoid release influences the inflammatory response to Covid-19“.

2. Lines 74 to 75 (reference 11) need to expand the mechanism of immune suppression in sepsis and the role of lipid mediators in the same. Maybe in brief.
Response: We thank the reviewer for this suggestion and we include a new phrase and reference: “In several sepsis models, it has been observed that specific CB2R agonists reduce leukocyte recruitment, oxidative burst, systemic inflammatory mediator production, bacteremia, and lung tissue damage, while enhancing survival {DOI: 10.1080/17843286.2018.1461754}”

3. Lipid regulation of viral replication should be supported with better references in addition to 21. (lines 94-96).
Response: We have revised this point accordingly and added new references (
Cannabinoid Receptor 2: A Potential Novel Therapeutic Target for Sepsis? Acta Clinica Belgica (doi:10.1080/17843286.2018.1461754).
Lipid droplets and lipid mediators in viral infection and immunity (doi: 10.1093/femsre/fuaa066), Bioactive lipids in antiviral immunity (DOI: 10.1126/science.abf3192)).

4. Role of glucocorticoids in severe COVID-19 is not controversial now. It has been proven to reduce mortality. So we should modify the line 97-98
Response: It is really a very relevant comment. Findings from the RECOVERY trial confirmed by the REMAP-CAP have revived interest in GCs as a current and effective standard-of-care treatment of severe SARS-CoV-2 pneumonia (COVID-19). However, there was not supported an advantage to using steroids in patients who are older or have more comorbidities. To make this clear, we added the statement in the text: "...the use of GCs in severe Covid-19 reduces mortality, but their risk/benefit balance remains inconclusive…”

5. This is cross-sectional study (line 108)
Response: We agree and we added this change in the text.

6. How many subjects of COVID-19 who were diagnosed on the basis of serological assays were classified to have severe disease or acute disease?
Response: In total, only 10 individuals with COVID-19 classified as mild/moderate performed the diagnostic based on serological assay, actually they were in home-care. Such information was added in the Supplementary Table 1.

7. Subjects on immunomodulators- were they included or excluded? In either case must be mentioned in the methods section.
Response: We have revised this point accordingly in the methods section.

8. Have the authors tried to evaluate the samples of same cases- before and after the administration of GC?
Response: As this is a cross-sectional study, we were unable to carry out this assessment on our participants, since we only selected one-time point samples without the possibility of follow-up.

9. Severity correlation with respect of pulmonary functions in the form of PaO2/FiO2 ratio can be added, if the authors have that data. This will have better correlation value as comapred to hematological parameters which might be affected due to medicaitons also in addition to the disease severity.
Response: We have used the PaO2/FiO2 ratio values as parameters to classify the COVID-19 severity according to the clinical score with a continuous classification variable. As suggested by the reviewer, we used this data to compare the influence on the inflammatory parameters in a correlation matrix; however, we did not observe any significant expressive changes. Also, we add the oxygen saturation (%) and PaO2/FiO2 ratio values to Supplementary Table 1.

Reviewer 2 Report

File.

Author Response

Response to Reviewer 2 Comments:

Reviewer #2:

The study was well conducted, with clearly specified aim and testable hypothesis.

The manuscripts’ introduction sums up well written information supported by up-to-date literature regarding the subject of the article, as well as highlighted issues, such as currently still lacking elucidation of molecular mechanisms underlying the immune response in COVID19 disease and still “controversial” GCs role in its treatment. Minor improvements are required. Methods and results are clearly textually and graphically presented, with results matching the described methodology.

Response: The authors thank Reviewer #2 for the positive evaluation of our manuscript.

Discussion also provides useful review of currently obtained knowledge in comparison to results of this study with significant novel information and contribution to the field. Authors highlighted the strengths and limitations of the study very well. Minor improvements are required.

  1. Line 36 – additional = additionally/ in addition.

Response: We have revised this point accordingly.

  1. Line 38 – Please provide a more precise term instead of lipid system (e.g. lipid pathways, lipid membranes, etc.).

Response: We have revised this point accordingly.

  1. Line 58 – Perhaps reformulate “levels produced” (e.g. produced amount, level of activation, increased activity, etc.).

Response: We have revised this point accordingly.  

  1. Line 86 – complicated = complex

Response: We have revised this point accordingly.

  1. Line 99-105 – This part should be moved to discussion regarding results/conclusion, not introduction

Response: We respected the reviewer opinion, but we consider that is a style of writing a research paper introduction: Map out your paper -The final part of the introduction is often dedicated to a brief overview of the rest of the paper, as suggested by UCI ( https://guides.lib.uci.edu/c.php?g=334338&p=2249903).

  1. Line 461-463 – Please reformulate/untangle this sentence, language-wise. Term “implications” is incorrectly used, perhaps severe cases, outcomes, etc.

Response: We have revised this point accordingly in the Discussion.

  1. Line 464 – Due of – due to

Response: We have revised this point accordingly in the Discussion.

  1. Line 519 – curiosity = curiously, oddly

Response: We have revised this point accordingly in the Discussion

  1. Line 524 – Please reformulate “produces inflammation”

Response: We have revised this point accordingly in the Discussion.

  1. Line 526 – reformulate to “effective eradication of the virus”

Response: We have revised this point accordingly in the Discussion

  1. Lines 539-541 – This is an oversimplification of the conclusions of RECOVERY study, either provide more details or delete this line and only mention the results of RECOVERY study relevant in comparison to your study (which you state in the next line).

Response: We thank the reviewer for this important suggestion and we include new statement.

  1. Lines 631-635 – How does this relate to your study and why is it mentioned in conclusion? To my understanding, this was not the main topic of the study, it should be stated either in the introduction or well written part of limitations of the study in discussion.

Response: We have revised this point accordingly.

  1. Line 635 – work established = reformulate to this study provided insight, elucidated

Response: We have revised this point accordingly in the Conclusions.

  1. Line 640 - To this end = reformulate (with the results/knowledge obtained in this study).

Response: We have revised this point accordingly in the Conclusions.

Reviewer 3 Report

This is an observational and descriptive study that was conducted in Brazil from June 108 2020 to January 2021, prior to the introduction of Covid-19 vaccination.

COVID-19 vaccination did not exist. However, the treatment guidelines with GCs were in place. In July 2020, the RECOVERY trial results and hence treatment guidelines were released (epub ahead of print).

So you must re-phrase your conclusion.

Figure 1. I cannot see the bars, please change the y-axis.

What GC dose did you use?

Table S1 is missing a lot of information, and mainly biochemical results.

Table with mild/moderate vs severe/critical must be shown.

Author Response

Response to Reviewer 3 Comments:

Reviewer #3:

This is an observational and descriptive study that was conducted in Brazil from June 2020 to January 2021, prior to the introduction of Covid-19 vaccination.

  1. COVID-19 vaccination did not exist. However, the treatment guidelines with GCs were in place. In July 2020, the RECOVERY trial results and hence treatment guidelines were released (epub ahead of print). So you must rephrase your conclusion.

Response: We thank the reviewer for this observation and we include new statement in Conclusions.

  1. Figure 1. I cannot see the bars, please change the y-axis.

Response: We performed a tracking break on the Y-axis for better display of results. It is really a very relevant comment.

  1. What GC dose did you use?

Response: It is a very relevant comment. The hospitalized patients in our cohort were GC-treated according to RECOVERY protocol: oral or intravenous dexamethasone (at a dose of 6 mg once daily) for up to 10 days (or until hospital discharge if sooner) and corresponding to their clinical status. This information was added to Material and Methods.

  1. Table S1 is missing a lot of information, and mainly biochemical results.

Response: We appreciate the reviewer's proposal and have added new clinical, laboratory, and biochemical information to Table S1. However, a lack of homogeneity in the clinical laboratory testing utilized by the hospitals participating in this study, for the care of patients with Covid-19, hindered the accuracy of additional laboratory and clinical data. The Discussion section has been updated to include this restriction.

  1. Table with mild/moderate vs severe/critical must be shown.

Response: We have revised this point accordingly.

Reviewer 4 Report

This is a very interesting paper on the effect of GCs on lipid pathways. However, I have some major concerns.

First of all, your conlusion:

"There are currently no data regarding the safety and efficacy of GCs in the treatment of patients with early-stage Covid-19 who are not hospitalized. Numerous clinical investigations indicate, however, that if the treatment begins too early, in the absence of severe symptoms, or too late, in patients with multi-organ failure, there is no benefit or perhaps a worse prognosis [101]."

RECOVERY TRIAL!!!!! treatment with dexamethasone at a dose of 6 mg once daily for up to 10 days reduces 28-day mortality in patients with Covid-19 who are receiving respiratory support. We found no benefit (and the possibility of harm) among patients who did not require oxygen).

Figure 1:  Please use log for y-axis as all scatter plots are at the 0 line and the differences cannot be seen.

You have used 24-hour admission blood. What about outpatients? ICU patients, admission to the ICU or the hospital?

Table S1: What is the GC dose for the oupatients? I gather that outpatients and ICU patients did not receive the same dose.

Severe/critical: How many were ICU patients? Mechanically ventilated?

Why did you pool these 2 groups together? There are a lot of differences between severe (eg high flow patients) and ICU intubated patients.

Figure 3a: I want to see the p-values. Or use *p<0.05, **p<0.01, ***p<0.001, ****p<0.0001

Table S1: Divide patients into mild/moderate and severe/critical. i.e. the three groups you studied.

So 53 subjects were outpatients? Please specify in your study population.

Strengths: You mention that "Our study distinguishes itself from others since it comprises a bigger group of individuals who have been clinically described in depth". Not really. Table S1 missing a lot of clinical information. PaO2/FiO2, d-dimers, CRP, PCT, LDH and many biochemical markers.

Table S2 and S3. Please use up to 3 decimal places.

Author Response

Response to Reviewer 4 Comments:

Reviewer #4:

This is a very interesting paper on the effect of GCs on lipid pathways. However, I have some major concerns.

First of all, your conclusion:

"There are currently no data regarding the safety and efficacy of GCs in the treatment of patients with early-stage Covid-19 who are not hospitalized. Numerous clinical investigations indicate, however, that if the treatment begins too early, in the absence of severe symptoms, or too late, in patients with multi-organ failure, there is no benefit or perhaps a worse prognosis [101]."

RECOVERY TRIAL!!!!! treatment with dexamethasone at a dose of 6 mg once daily for up to 10 days reduces 28-day mortality in patients with Covid-19 who are receiving respiratory support. We found no benefit (and the possibility of harm) among patients who did not require oxygen).

Response: We thank the reviewer for this observation and we include new statement in Conclusions.

  1. Figure 1: Please use log for y-axis as all scatter plots are at the 0 line and the differences cannot be seen.

Response: We have revised this point accordingly and performed the tracking break on the Y-axis for better display of results.

  1. You have used 24-hour admission blood. What about outpatients? ICU patients, admission to the ICU or the hospital?

Response: We apologize for not being clear enough. Peripheral blood samples were collected within the first 24 hours of patient enrollment to the study, but not 24 hours after hospital admission. We have revised this point accordingly in Material and Methods. 

  1. Table S1: What is the GC dose for the oupatients? I gather that outpatients and ICU patients did not receive the same dose.

Response: It is a very relevant comment. The hospitalized patients in our cohort were GC-treated according to RECOVERY protocol: oral or intravenous dexamethasone (at a dose of 6 mg once daily) for up to 10 days (or until hospital discharge if sooner) and corresponding to their clinical status The outpatients did not receive GC treatment. This information was added to Material and Methods.

  1. Severe/critical: How many were ICU patients? Mechanically ventilated?

Response: We thank the reviewer for this important question and we have revised this point accordingly with additional information in Table S1.

  1. Why did you pool these 2 groups together? There are a lot of differences between severe (eg high flow patients) and ICU intubated patients.

Response: The referee is right about clinical differences between severe and critical patients. However, since our observations are mostly related to the pharmacological treatment regimen of the patients, especially by GC, we decided to include severe and critical patients in the same study group due to these similarities in treatment and hospital care.

  1. Figure 3a: I want to see the p-values. Or use *p<0.05, **p<0.01, ***p<0.001, ****p<0.0001.

Response: As suggested by the reviewer, we made the comparisons between these groups and showed the fold change (FC), the nominal p-value, and the FDR adjusted p-value in Supplementary Table 2, referring to the differential gene expression and their respective p-values. Also, we included a statement in the figure 3 legend: “Significant differences in transcript expression was accessed using Benjamini and Hockberg adjusted p-values to controlling false discovery rate (FDR) obtained from whole transcriptome differential expression analysis considering a threshold of FDR < 0.05. Table S2 contained gene fold change values, nominal and FDR adjusted p-values obtained from the differential expression analysis between the analyzed groups”.

  1. Table S1: Divide patients into mild/moderate and severe/critical. i.e. the three groups you studied.

Response: We thank the reviewer for this suggestion and we have revised this point accordingly.

  1. So 53 subjects were outpatients? Please specify in your study population.

Response: In fact, all the mild and most of the moderate Covid-19 patients were in home-care, which was recommended for the management of patients in that pandemic situation. However, 10 patients of the 53 subjects in this mild/moderate clinical group required hospitalization due to previous comorbidities and the risk of Covid-19.

  1. Strengths: You mention that "Our study distinguishes itself from others since it comprises a bigger group of individuals who have been clinically described in depth". Not really. Table S1 missing a lot of clinical information. PaO2/FiO2, d-dimers, CRP, PCT, LDH and many biochemical markers.

Response: We appreciate the reviewer's proposal and have added new clinical, laboratory, and biochemical information to Table S1. However, a lack of homogeneity in the clinical laboratory testing utilized by the hospitals participating in this study, for the care of patients with Covid-19, hindered the accuracy of additional laboratory and clinical data. The Discussion section has been updated to include this restriction.

  1. Table S2 and S3. Please use up to 3 decimal places.

Response: We thank the reviewer for this suggestion and we have revised this point accordingly.

Round 2

Reviewer 3 Report

No further comments